# Exploring the Role of Community Empowerment in Urban Poverty Eradication in Kuala Lumpur, Malaysia

Mahaganapathy Dass [1], Puvaneswaran Kunasekaran [2,*], Charanjit Kaur [3] and Sarjit S. Gill [2,*]

1 Lim Kok Wing University of Creative Technology, Cyberjaya 63000, Malaysia
2 Faculty of Human Ecology, University Putra Malaysia, Serdang 43400, Malaysia
3 Faculty of Creative Industries, Universiti Tunku Abdul Rahman (UTAR), Kampar 31900, Malaysia
* Correspondence: puvaneswarankunasekaran@gmail.com (P.K.); sarjit@upm.edu.my (S.S.G.);
  Tel.: +60-122-591-410 (S.S.G.)

**Abstract:** The main purpose of this study was to holistically understand the role of empowerment in urban poverty eradication of the Indian community in the urban areas of the Klang Valley, Malaysia. The poverty eradication effectiveness was tested by analysing community empowerment domains and MyKasih programme run by a non-governmental organisation. There are numerous studies conducted to understand the issues of poverty in Malaysia. However, few studies have so far focused on the minority community in Malaysia. Moreover, there is no recent study to test the effectiveness of any governmental or NGO's poverty eradication efforts on this minority community. This study utilised a qualitative approach and an in-depth interview was used to gather the data. The respondents were single mothers living in a poverty-stricken area in the capital city of Malaysia.

**Keywords:** urban poverty; minority community; empowerment; community development





## 1. Introduction

According to the Malaysian urban poverty statistics, almost 80 percent of the country's population consists of Indians [1]. Unregulated labour standards, poor working conditions, and an absence of job security all contribute to this issue. In contrast, the Malay community in rural Malaysia is disproportionately impoverished. Low levels of education, low skills, low income, low work status, and poor housing in rural areas with inadequate basic amenities all contribute to this issue. Consequently, this event has a detrimental effect on them, leading them to be anxious and to fight every day to make ends meet [2].

A large number of foreign workers undoubtedly exacerbates the urban poverty problem [3]. Consequently, migrant labourers from other countries have become an additional cause of poverty. As a result, job possibilities have a direct effect on the urban poor, leading to low income and unemployment [4]. As Malaysia is a rapidly expanding nation, there is a significant demand for qualified and skilled labour. Consequently, an increase in the foreign labour force is contributing to Malaysia's economic growth [5].

Due to work uncertainty, urban Indians face significant financial hardship; permanent positions in the manufacturing and construction industries can help them attain job stability and a better salary, allowing them to progress socially and economically. In contrast, the construction sector commonly hires temporary or day-to-day labour, making them economically vulnerable [1]. This challenge is clear when a significant number of these Indians work in dead-end professions, with no promotion opportunities and serious occupational and health dangers.

The MyKasih programme was selected for an effectiveness evaluation because it focuses on alleviating poverty among low-income urban residents. The MyKasih programme, started in April 2009, is a non-profit organisation that offers low-income families food aid and educational opportunities [6]. The project is supported by the Selangor Pilgrims Funds Board (Lembaga Zakat Selangor), the Johor State Government, and the Ministry of Federal

Territories and Urban Well-being. Urban locations such as Lembah Pantai, Selayang, Batu, Wangsa Maju, Petaling Jaya, and Pasir Gudang are the focus of the initiatives [6]. The primary objective of the MyKasih food aid and student bursary programme is to assist underprivileged families in putting food on the table and keeping their children in school, with the ultimate objective of eradicating poverty. MyKasih also seeks to provide for a family's basic needs, ensuring that their children receive a quality education and have access to improved employment possibilities.

"Our mission is to assist poor and needy families in achieving more independence and to offer their children an opportunity to escape the cycle of poverty. We do not randomly provide food to the poor and disadvantaged. Instead, we are committed to the long-term empowerment of those we serve. In addition to allowing children to make their own food purchases, we teach them essential life skills. According to the MyKasih Foundation, many impoverished individuals live in dilapidated dwellings with little chance of escaping poverty. We experienced these problems personally, and it became evident that action was required. We quickly realised that merely providing poor families with food, money, and clothing was insufficient. Frequently, nutritional needs are not satisfied, health standards degrade, and children's schooling drops, resulting in diminished job opportunities. The cycle of misfortune and destitution begins anew" (Informal Communication, 20 February 2020, the 20th Sukhbindar Singh, Project Manager, MyKasih Foundation).

In addition to food assistance and student bursaries, MyKasih conducts health screening workshops, financial literacy and skills training courses, tuition classes, and income-generating initiatives to help beneficiaries achieve financial independence by providing opportunities to enhance their economic productivity and quality of life. Additionally, the need to stress the social development of communities is also highlighted in the sustainable development goals. The goals were initially environmentally focused, during the Brundland Commission (1987), but gradually created more emphasis on the social aspects in SDG17.

## 2. Materials and Methods

By 2025, the majority of people in the Southeast Asian area will live in urban slums due to the region's continued urbanisation. This can be seen in the demography and population growth of urbanised cultures and nations such as Japan, China, and India [7]. Aside from that, East and Southeast Asia are anticipated to reach the 50% level by 2015, while South Asia might achieve this same rate by 2030. Between rural and urban poverty, there are notable distinctions. Living in substandard circumstances, not having enough to eat, not having access to education, and not having a job are all considered to be signs of poverty in Malaysia.

According to the United Nations Economic and Social Commission for Asia and the Pacific, absolute poverty is the cost of the bare necessities required to sustain human life, while relative poverty is the cost of the bare necessities required to maintain an acceptable standard of living in a particular society. Furthermore, the European Union defines the relatively poor as individuals, families, and groups of individuals whose access to material, cultural, and social resources prevent them from enjoying the bare necessities of life in the member state in which they reside.

The scenario which combines individual strengths and talents, natural healing processes, and proactive behaviour in social policy and social change-related themes [8] can be defined as empowerment. In addition, the capacity of individuals, groups, or organisations to implement something by themselves can be also classified as empowerment. It is a measurement of a product or service's usefulness to others [9]. In addition, empowerment is the chance and the capacity to make decisions that directly impact consumers [10]. One of the most debated components of community development is empowerment. To accomplish change, community life requires the distribution of decision-making authority. As a method of managing the vital force, it is entitled to democratic participation [11].

There are three dimensions of empowerment: (1) the development of a more positive feeling and strong sense of self; (2) the construction of knowledge and the capacity to more critically comprehend the social and political realities of the network; and (3) the cultivation of resources and strategies, or more functional competency, to achieve personal and collective goals [12]. Numerous critiques of the problematic notion of empowerment have been published, and these studies imply that modern researchers should adopt a more critical approach to the term and be more precise about the techniques that they contend have promoted empowerment [13].

An inclusive approach to such assessments requires a focus on gender and other inequities, as well as the utilisation of participatory evaluation processes [14]. Moreover, a number of researchers argue that it is necessary to develop effective models of empowerment and methods for analysing and critically evaluating claims of empowerment [15]. Thus, it can be restated that communities have access to three types of power: social, political, and psychological, and that social power involves access to specific "bases" of household products, such as information, knowledge, and skills, as well as participation in social organisations and financial resources. This type of empowerment comprises key features of social capital, which has been identified as a critical element of sustained community development [16].

The idea of "political power" comprises access to the process by which choices are made, especially those that impact people's personal destinies, as well as the right to vote, the right to free expression, and the right to collective action [14]. Experts believe that social empowerment must precede meaningful participation.

However, the core Zimmerman theory of empowerment argues that such agents may play a role in offering assistance in a way that helps the disempowered to abandon conventional reliance. Although somewhat idealistic, Friedmann's empowerment framework is useful because it suggests that empowerment and social change is a multidimensional process requiring analysis at the micro and macro levels of the individual and the community, organisation, or group, as well as their interrelationships.

The empowerment process involves offering the opportunity to acquire the skills required to influence others, engage in organisational decision making, and take action to achieve social change [17]. Empowerment is a dynamic process including the growth of choice and action freedom, as well as the ability to influence behaviour and social organisation [18]. Due to our propensity for luxury, this goal is difficult to attain. Empowerment via participation is a continual process through which individuals build and utilise their potential to take action on shared problems and achieve the expected conditions in order to enhance the lives of others [19]. Thus, it can be generally argued that empowerment can create a sense of self-reliance for the community without being too dependent on outsiders, especially government and NGO assistance. By being empowered, the urban poverty issue can be eradicated in the long term in a sustainable manner.

In this study, the qualitative method was used to collect data from respondents. An in-depth interview approach was used to acquire data. The respondents were Malaysian single mothers living in low-cost housing complexes. Theoretical saturation was utilised to determine the sample size. In this case, nine respondents were interviewed before reaching a saturation point. Snowball sampling was utilised to determine who was to be interviewed next.

The Klang Valley was chosen as the research's subject location. Because it is the most established and developed region in the nation, this location was chosen. Klang Valley is approximately 50 km long and 25 km wide. Subang, Petaling Jaya, Shah Alam, and Klang are the initial important cities in this area [20]. Putrajaya, Cyberjaya, Bangi, and Ampang are just a few of the new cities that have been built as a result of urbanisation. The Klang Valley is regarded as Malaysia's geographic core [21]. The nation's industrial and commercial centre is located in this area. There were 4.7 million people living in the Klang Valley overall as of the year 2006. There was a total population of 8 million people by 2021, thanks, in part, to the temporary foreign workers who have moved to the area [22].

Indians relocate often from rural and suburban areas to the Klang Valley, which is teeming with amenities and employment prospects. However, for this study, a specific region of the Klang Valley was chosen as the sample area. Due to its advantageous location as an urban region for migration from rural areas, Lembah Pantai was chosen. In the Klang Valley, this location has one of the highest concentrations of Indians.

The names and identities of the key respondents were not revealed to obtain genuine and neutral responses. All of the respondents were from the Lembah Pantai government low-cost apartment area in Kuala Lumpur as mentioned in Table 1. In addition, many key respondents also requested not to reveal their names as they were still dependent on the MyKasih organisation, and if they said anything against their sponsors, they would be disqualified from any assistance.

**Table 1.** Details of Key Respondents.

| Name of Respondent | Number of Children | Employment | Type of Home |
|---|---|---|---|
| Respondent 1 Female, 45 years old, Divorced | Five children, living together with mother | Food stall operator, 12 h/per day | Rental at government apartment |
| Respondent 2 Female, 38 years old, Divorced | Two children, living together with their mother | Hairstylist, only weekends | Rental at government apartment |
| Respondent 3 Female, 51 years old, Divorced | Four children, two living together with their mother and two more left the house | Bakery at home, 8 h/per day | Rental at government apartment |
| Respondent 4 Female, 48 years old, Separated | Five children, living together with their mother | Food stall operator, 12 h/per day | Rental at government apartment |
| Respondent 5 27 years old, Female, Divorced | One child, living together with their mother | Make-up artist, only weekends | Rental at government apartment |
| Respondent 6 Female, 32 years old, Divorced | Four children, living together with mother | Manicure and pedicure operator, only weekends | Rental at government apartment |
| Respondent 7 Female, 41 years old, Divorced | Seven children, living together with mother | Tailor, 12 h/per day | Rental at government apartment |
| Respondent 8 Female, 50 years old, Divorced | Nine children, five living together with mother and four left house | Food stall operator, 12 h/per day | Rental at government apartment |
| Respondent 9 Female, 55 years old, Divorced | Two children, one living together with their mother and one passed away | Food stall operator, 12 h/per day | Rental at government apartment |

More than half of the respondents claimed that they did not have any knowledge of financial management. One of the main reasons for this, stated by the respondents, was that they were not educated and could not afford to attend courses on financial management, due to poverty. The other reason was ignorance.

## 3. Results

The outcome of the study was segregated into the economic and political empowerment themes. These two themes were found to be dominant in the study.

### 3.1. Economic Empowerment

The majority of the respondents stated they could run their own businesses without assistance from others. They stated that they would rather be a business owner than an employee. In addition, they asserted that they were capable of conducting evaluations and making recommendations to enhance the MyKasih entrepreneurship programme.

*"I would rather become a business owner than an employee, and I am convinced that I can operate my own firm without outside support. I am capable of doing an evaluation and making suggestions to enhance the MyKasih Entrepreneurship Programme. I am confident that by participating in the MyKasih Entrepreneurship Programme, I would be*

*able to successfully operate my own business without outside assistance. Nevertheless, I am not yet prepared to evaluate the MyKasih Entrepreneurship Programme or make recommendations for its enhancement. Now that I have greater comprehension, knowledge, and abilities, I can make independent decisions. I want to be an entrepreneur instead of an employee; but, I am unable to conduct these evaluations or make recommendations. Since I can now make independent decisions, I am certain that I can operate my own firm profitably. Moreover, once I become a business owner, I do not believe I can return to being an employee. Since I lack appropriate knowledge and expertise, I do not believe I am equipped to evaluate the MyKasih Programme or give suggestions for its enhancement."* (Respondent 1, 45 years old, from Kuala Lumpur)

*"As I stated before, I am confident that my business understanding, knowledge, and abilities have increased, and as a result, I am able to make autonomous judgments, especially about the management of my firm, and I have no intention of turning back. I will always be a business owner, never again an employee. I am unable to perform assessments or give suggestions for enhancing the MyKasih Programme because I lack the relevant knowledge and experience. My business-related understanding, knowledge, and skills have grown, and as a result, I am now confident enough to make independent business judgments. After this, I no longer believe I could be an employee. It should be an entrepreneur. I am unable to conduct assessments or make recommendations for change since I lack the necessary qualifications."* (Respondent 6, 32 years old, from Kuala Lumpur)

Nonetheless, some respondents stated that they lacked the necessary understanding, expertise, and abilities to manage a firm, and are, therefore, not yet prepared to become entrepreneurs. Similarly, they were incapable of conducting any type of evaluation or making suggestions for change.

*"Since I did not gain much from this programme and my business understanding, expertise, and abilities have not increased significantly, I cannot be a good business owner since I cannot make quick decisions. I will not be able to complete evaluations or provide improvement suggestions for the MyKasih Entrepreneur Programme."* (Respondent 7, 41 years old, from Kuala Lumpur)

*"I am unsure about my ability to handle a business on my own because I did not learn anything from them. I am aware that I am unable to make independent business decisions since my business understanding, knowledge, and abilities are insufficiently developed. I am unable to conduct evaluations or give suggestions for improving the MyKasih Entrepreneur Programme."* (Respondent 8, 50 years old, from Kuala Lumpur)

The output of an empowerment process is to provide a platform for self-evaluation, independent management capacity, and proactive action as a preventative step against future inadequacies. The moment a community empowerment strategy is implemented, a revolution in terms of economic prosperity and capacity building is possible.

*3.2. Political Participation*

The majority of respondents stated that they would prefer be leaders than followers owing to their impoverished backgrounds, which would make them excellent leaders. They argued that they could not survive without government assistance, since they required financial resources until they could support themselves. In addition, they stated that they cannot lead the community without assistance from the outside.

*"I would rather be a leader than a follower, but in order to be a competent leader, we need appropriate resources, especially financial support from the government. I can run my household without assistance, but what about the community? I cannot serve as community leader without aid from the government or third parties. I am convinced that I would make an excellent leader because I am familiar with the community's needs and desires. Nonetheless, leaders have their own families; who will provide for them?*

*This implies that leaders must have financial assistance, which must come from outside sources such as the government if they are not financially independent. Since we are impoverished, the government must provide the necessary resources. "Who will take care of us if the government does not?"* (Respondent 2, 38 years old, from Kuala Lumpur)

In general, respondents believed they could lead their families and provide them with a normal existence without assistance from the outside, but they could not do the same for the society. They required government assistance to lead the community because it was difficult to operate without finances.

The study's findings are divided into economic and political empowerment topics. In the research, these two motifs were shown to be dominant.

### 3.3. Economic Autonomy

The majority of respondents indicated that they could operate their own firm independently. They expressed a preference for company ownership over employment. In addition, they declared that they are competent to perform assessments and make suggestions for improving the MyKasih entrepreneurship programme.

*"I'd rather be a business owner than an employee, and I'm certain that I can run my own company without assistance. I am capable of evaluating and recommending improvements to the MyKasih Entrepreneurship Programme. I am certain that through enrolling in the MyKasih Entrepreneurship Programme, I would be able to manage my own business independently with success. However, I am not yet able to evaluate the MyKasih Entrepreneurship Programme or give suggestions for its improvement. Now that I have increased insight, knowledge, and skills, I am able to make judgments independently. I would rather be an entrepreneur than an employee, yet I am unable to do these assessments or offer these suggestions. Since I can now make my own judgments, I am certain that I can run my own business profitably. Moreover, once I become a business owner, I do not believe I would be able to return to the employee status. Since I lack the necessary knowledge and skills, I do not feel qualified to evaluate the MyKasih Programme or make recommendations for its improvement."* (Respondent 1, 45 years old, from Kuala Lumpur)

*"As I've previously indicated, I am convinced that my business understanding, expertise, and talents have developed, and as a consequence, I am competent to make independent decisions, particularly regarding the administration of my company, and I have no intention of going back. I will never again be an employee; I will forever be a business owner. Due to my lack of relevant expertise and experience, I am unable to conduct assessments or provide ideas for improving the MyKasih Program. My business-related information, expertise, and confidence have increased, and as a result, I am now able to make autonomous business decisions. After this, I no longer consider myself employable. It should be a businessperson. Since I lack the required credentials, I am unable to conduct evaluations or offer recommendations for change."* (Respondent 6, 32 years old, from Kuala Lumpur)

Nonetheless, several respondents said that they lack the knowledge, skills, and experience required to operate a business and are therefore not yet prepared to become entrepreneurs. Likewise, they are incapable of undertaking any form of review or proposing changes.

*"Because I did not acquire much from this programme and because my business knowledge, competence, and skills have not considerably expanded, I cannot be a competent business owner because I cannot make rapid judgments. I will not be able to complete MyKasih Entrepreneur Programme assessments or make recommendations for enhancements."* (Respondent 7, 41 years old, from Kuala Lumpur)

*"I am uncertain about my capacity to manage a firm on my own because I gained no knowledge from them. I am conscious that I am unable to make autonomous business*

*judgments due to my lack of business understanding, expertise, and skills. I am unable to evaluate the MyKasih Entrepreneur Programme or provide ideas for its improvement."* (Respondent 8, 50 years old, from Kuala Lumpur)

The outcome of an empowerment process is a platform for self-evaluation, autonomous managerial capability, and proactive action as a precaution against future deficiencies. As soon as a plan for community empowerment is adopted, a revolution in terms of economic growth and capacity building is feasible.

*3.4. Political Participation*

Due to their disadvantaged backgrounds, the majority of respondents claimed that they would rather be leaders than followers, which would make them outstanding leaders. They stated that without government aid, they could not exist, since they lacked the financial resources until they could sustain themselves. Additionally, they asserted that they could not lead the community without outside support.

*"I'd rather be a leader than a follower, but in order to be a successful leader, we need the proper resources, particularly government funding. My home is self-sufficient, but what about the community? I cannot act as a community leader without government or third-party assistance. I am confident that I would be a good leader since I am familiar with the needs and desires of the community. Still, leaders have their own families; who will support them? This means that leaders must have financial support from outside sources, such as the government if they are not financially independent. The government must supply the required resources due to our poverty. Who will care for us if the government fails to do so?"* (Respondent 2, 38 years old, from Kuala Lumpur)

In general, respondents felt they could provide their family with normal living without outside support, but they could not do the same for society. It was difficult for them to lead the community without government aid, as it was tough to function without funds. The Friedmann community empowerment theory is the key theory that informed this study's discussion of eradicating urban poverty. Moving from a rural to an urban area, or vice versa, is no longer the primary factor boosting socio-economic welfare [23]. The study's findings also did not conflict with Arnstein's ladder of participation theory, which claims that the community is either coerced or persuaded to take part in MyKasih programmes, preventing true community participation. Numerous members still rely heavily on MyKasih's assistance and state intervention. This verifies the argument that political sociology and wages have had an impact on the culture of poverty theory [24].

The researchers found that not all of the residents were motivated by the knowledge gained and skills learnt from their participation in the MyKasih entrepreneurship programme, simply because they were content with their poor lifestyle and they were not willing to improve their livelihood in their settlement. Most of them believed that they were helpless and dependent on the government for assistance. They also suffered from an inferiority complex, due to a lack of education and employment opportunities.

The findings of this research demonstrated that community people may participate in all facets of life through empowerment, especially while a development programme is in progress. This is because the public would be involved in the program's empowerment. In order for people to determine whether or not to engage, they need to be made aware of a genuine problem [25]. This is in line with the viewpoint stressing that political activism is actually a kind of empowerment [19]. The power to enact social and economic justice, as well as the spirit of democracy, lies in the people's participation [26]. As a result, participation is a means of obtaining the rights of the people. The process of decision making involved in community empowerment requires the consideration of the interests of the community itself. Thus, the contribution of this study can be seen from the theoretical and practical perspectives.

## 4. Conclusions

The conclusion of the study does not contradict the participation hypothesis, which claims that true community engagement in MyKasih programmes cannot occur if the community is coerced or induced to join. The study's findings also do not contradict the empowerment idea, which states that empowered individuals will be free of poverty. Indians, who participated directly in MyKasih initiatives, tended to have good views of self-development, since they obtained tangible economic advantages through small business and employment participation. Although the results do not significantly contradict the culture of poverty hypothesis, which claims that the socioeconomic wellbeing of a poor urban population is influenced by the systematic application of government policies and processes, they do not support the idea. Within the context of community empowerment, the findings of this study can be seen as an essential branch of minority poverty eradication through empowerment.

The perception of the community is often neglected by the authorities, making the community feel that they are not empowered [26,27]. The community agreed that the outsiders, especially the government, always discuss with the community before the implementation of any projects [28,29]. According to the respondents, the discussions are considered as mere routine for the officers to show respect to the residents. The final decision on development projects will eventually be made by the government. If this continues, the community will feel detached from any community development projects initiated by the government. Thus, the government should not just consult the community simply to meet the administrative procedures [30–32]. They should respect the community's ideas and give them opportunities to make decisions [33]. Through doing this, the community can learn by themselves and improve their status.

**Author Contributions:** Conceptualisation, M.D. and S.S.G.; methodology, P.K.; writing—review and editing, C.K. All authors have read and agreed to the published version of the manuscript.

**Funding:** This research received no external funding.

**Conflicts of Interest:** The authors declare no conflict of interest.

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
