# Peer review of "Exploring the Role of Community Empowerment in Urban Poverty Eradication in Kuala Lumpur, Malaysia"

_sustainability, doi:10.3390/su141811501_

Round 1

Reviewer 1 Report

The topic of the article seems to me to be of great interest, however the way of carrying out the analysis I think could be improved.

I'm going to make some comments:

1-It would be necessary to highlight more the social interest of the study

2-It would be necessary to highlight the contribution of the work

3-I think that the links between the two fundamental aspects discussed in the article are confused or at least poorly explained: eradication of poverty and empowerment.

4-I think that a greater explanation of the applied methodology would be needed: in-depth interviews carried out and the qualitative method.

5-The conclusions should be extended, since they are very brief

Author Response

First of all, thank you for providing us with the opportunity to submit revised version of our manuscript. We greatly appreciate the encouraging and thought-provoking comments given in response to our manuscript submission. We have addressed the reviewers’ comments, and the attached document shows the comments and the corrections incorporated into the revised manuscript by the authors. We would once again like to thank the reviewers for their detailed and helpful suggestions. We hope that our revision satisfies you and the reviewers, and that our paper will now meet the standards of the special issue.

Best regards,

The Authors

Reviewer 2 Report

The objective of this paper is to holistically understand the role of empowerment in urban poverty eradication of the Indian community in the urban areas of Klang Valley, Malaysia. This is an interesting topic, and the paper is fairly well written, though there are a few grammatical and typographical errors.

The paper investigates an important and timely topic. However, the study appears to be based on a very small number of semi-structured interviews. Authors have not clearly stated how the sample of participants were selected and how many participated in the study. This is a main weakness of the study. Though the paper is a qualitative study, authors could have presented some type of summary statistics. Instead of focusing only on the opinions of some people who have been interviewed, it will add value to the paper if some additional analyses are conducted.

The methodology of the paper also not discussed well. Conducting interviews and collecting primary data is appropriate for a study of this nature. One of the main weaknesses of the study is limiting it to the opinions of a few participants. Nevertheless, the conclusions adequately tie together the other elements of the paper. In addition, authors have also compared the findings of this study with that of previous studies.

Author Response

Response to Reviewer 2 Comments

Point 1:  Authors have not clearly stated how the sample of participants were selected and how many participated in the study. This is the main weakness of the study. Though the paper is a qualitative study, the authors could have presented some type of summary statistics. Instead of focusing only on the opinions of some people who have been interviewed, it will add value to the paper if some additional analyses are conducted.

Response 1: Thanks for the comment. We have added more details about the respondents involved in the in-depth interview. Due to the word limit, only several opinions are projected. As this is a purely qualitative piece of work, we did not include any statistical output. We will definitely consider this in future. This is a valid point. Page 4, line 160-174.

Point 2: The methodology of the paper also not discussed well. Conducting interviews and collecting primary data is appropriate for a study of this nature. One of the main weaknesses of the study is limiting it to the opinions of a few participants. Nevertheless, the conclusions adequately tie together the other elements of the paper. In addition, authors have also compared the findings of this study with that of previous studies.

Response 2: Thanks for the comment. We have highlighted the contribution in page 3, lines 140-145.

Round 2

Reviewer 2 Report

The revised version of the manuscript shows some improvement over the original version. Authors have addressed my concerns about the previous version. However, a research based on the opinions of only nine participants still remains a weakness. In Table 1, the terms Informant1, Informant2, etc. can be changed to Respondent1, Respondent2, etc.

Author Response

Response to Reviewer 2 Comments

Point 1:  The revised version of the manuscript shows some improvement over the original version. Authors have addressed my concerns about the previous version. However, a research based on the opinions of only nine participants still remains a weakness. In Table 1, the terms Informant1, Informant2, etc. can be changed to Respondent1, Respondent2, etc.

Response 1: Thanks for the comment. We have changed ‘informants’ to ‘respondents’. Regarding the number of respondents, a limited number of nine respondents is due to data saturation. During the data collection, the researchers reached data saturation at the 9th respondent. Thus, the interview is stopped at the 9th respondent. We have added the explanation in page 3, lines 140-145.